# OTTER: A Vision-Language-Action Model with Text-Aware Visual Feature Extraction

**Huang Huang** [* 1]  **Fangchen Liu** [* 1]  **Letian Fu** [* 1]  **Tingfan Wu** [2]  **Mustafa Mukadam** [2]
**Jitendra Malik** [1 2]  **Ken Goldberg** [1]  **Pieter Abbeel** [1]

## Abstract

Vision-Language-Action (VLA) models aim to predict robotic actions based on visual observations and language instructions. Existing approaches require fine-tuning pre-trained vision-language models (VLMs) as visual and language features are independently fed into downstream policies, degrading the pre-trained semantic alignment. We propose OTTER, a novel VLA architecture that leverages these existing alignments through explicit, text-aware visual feature extraction. Instead of processing all visual features, OTTER selectively extracts and passes only task-relevant visual features that are semantically aligned with the language instruction to the policy transformer. This allows OTTER to keep the pre-trained vision-language encoders frozen. Thereby, OTTER preserves and utilizes the rich semantic understanding learned from large-scale pre-training, enabling strong zero-shot generalization capabilities. In simulation and real-world experiments, OTTER significantly outperforms existing VLA models, demonstrating strong zero-shot generalization to novel objects and environments. Video, code, checkpoints, and dataset: `https://ottervla.github.io/`.

## 1. Introduction

Recent advancements in Large Language Models (LLMs) and Vision-Language Models (VLMs) have inspired the exploration of scaling datasets and computational resources for vision-language-action (VLA) models (Collaboration et al., 2024; Khazatsky et al., 2024; Octo Model Team et al., 2024;

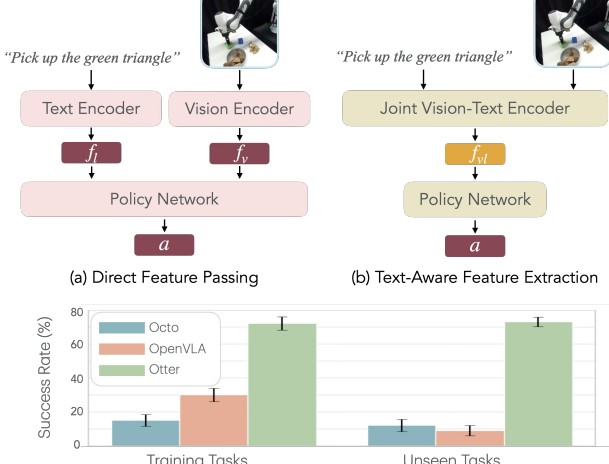

**Figure 1:** *(Top)* Different feature extraction approaches in VLA models. (a) Direct Feature Passing: existing approaches, exemplified by Octo and OpenVLA, extract and pass visual and text tokens independently to the policy network. (b) Text-Aware Feature Extraction: the proposed approach, OTTER, extracts visual tokens that correspond to the text tokens, and then feeds them into the policy. *(Bottom)* Real-world Robot Experiments: OTTER demonstrates higher success rates on both training and unseen real-world robot pick-and-place tasks compared to Octo and OpenVLA. OTTER exhibits better zero-shot generalization to unseen objects, maintaining strong performance across a variety of novel tasks.

Kim et al., 2024). Different input modalities are usually encoded into separate tokens: multi-view images encoded via visual feature extractors, along with tokenized language instructions, optionally with the robot's proprioceptive states, are fed into a transformer-based robot policy for end-to-end action generalization. This approach requires the policy network to connect the vision and language information and conduct precise robot control, which often presents significant challenges, especially in unseen environments.

Existing works such as RT-2 (Brohan et al., 2023) and Open-VLA (Kim et al., 2024) have demonstrated the benefits of directly fine-tuning pre-trained VLMs on robotic datasets to map vision and language to control. While these approaches leverage rich visual and language features from pre-trained encoders, this fine-tuning may interfere with pre-trained vision and language features, particularly since robot datasets are far less semantically diverse compared to large vision-language datasets (Schuhmann et al., 2022) where these

---

[*]Equal contribution [1]University of California, Berkeley [2]Meta AI. Correspondence to: Huang Huang <huangr@berkeley.edu>, Fangchen Liu <fangchen_liu@berkeley.edu>, Letian Fu <max.fu.letian@berkeley.edu>.

*Proceedings of the 42$^{nd}$ International Conference on Machine Learning*, Vancouver, Canada. PMLR 267, 2025. Copyright 2025 by the author(s).

(a) Multi-Modal State Representation

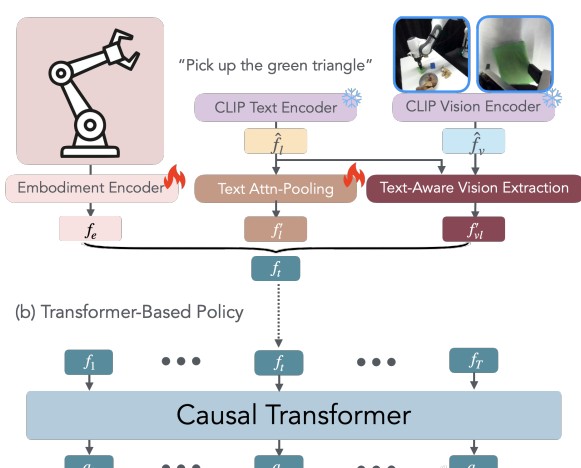

(b) Transformer-Based Policy

**Figure 2: OTTER Model architecture.** At each timestep $t$, text-aware visual features $f_{vl}$ are extracted from a pre-trained CLIP model (see Figure 3). Then $f_{vl}$ and the text tokens $f_l$ are further compressed through separate attention pooling layers, producing two representations $f'_{vl}$ and $f'_l$, respectively. The robot's proprioception is encoded by an MLP layer to generate the embodiment representation $f_e$. The tokens $f'_l$, $f'_{vl}$, and $f_e$ are then concatenated to form $f_t$, which serves as input to a causal transformer. With a context window of $T$ steps, the model autoregressively predicts the future actions ($a_t$) at each step.

VLMs are trained on, often leading to a noticeable drop in performance on unseen objects or environments compared to seen tasks (Brohan et al., 2023).

This finding is consistent with observations in vision-language research (Kerr et al., 2023; Lan et al., 2024), where fine-tuning language-aligned vision encoders, such as CLIP (Radford et al., 2021), has been shown to overfit and degrade generalization and long-tail classification performance. Yet, models like CLIP and SigLIP (Zhai et al., 2023) already exhibit strong vision-language alignment and impressive zero-shot performance on various downstream tasks, including fine-grained tasks like open-vocabulary segmentation (Rao et al., 2022; Lan et al., 2024; Dong et al., 2023). This raises the question of whether a more effective strategy is to extract and utilize the pre-trained alignment rather than risk weakening it through fine-tuning.

We seek to preserve the generalizability of VLMs for effective performance in unseen scenarios. To this end, we propose OTTER, a novel VLA architecture that freezes pre-trained vision and language encoders and extracts task-relevant visual features guided by language instructions. Instead of passing all visual features to the policy network, OTTER selectively extracts those semantically aligned with the task description. We use CLIP for its zero-shot capabilities and wide adoption, employing a lightweight selection mechanism to preserve its pre-trained vision-language understanding while adapting it for robotic control.

Figure 2 illustrates the architecture of OTTER, which in-

corporates text-aware feature extraction into the vision-language-action (VLA) pipeline. At each timestep, OTTER first processes visual and textual inputs to pinpoint task-relevant visual tokens. These selected features, along with proprioceptive data, enable the policy network to concentrate specifically on action planning. By keeping the pre-trained vision-language encoders frozen, OTTER effectively decouples task planning (selecting relevant visual features) from robot action planning (predicting appropriate actions). Both physical and simulation experiments demonstrate that OTTER outperforms existing VLA models, showing strong generalization to novel objects and environments with less performance degradation (Figure 1).

To summarize, our contributions are:

1. We propose OTTER, a VLA model that leverages the semantic alignment capabilities of pre-trained VLMs for better generalization. By extracting text-aware visual features that are semantically aligned with task instructions, OTTER preserves and utilizes the rich visual-language understanding from pre-training for effective robotic task execution.

2. OTTER significantly outperforms state-of-the-art VLA models on unseen robot manipulation tasks through its zero-shot generalization capabilities. By preserving the frozen pre-trained vision-language model rather than fine-tuning, OTTER effectively leverages the semantic understanding from large-scale pre-training to handle novel objects and environments.

3. Empirical results suggest that OTTER's performance on unseen tasks scales along multiple axes: through larger pre-trained vision-language encoders, increased policy network capacity, and pre-training on larger robot datasets.

## 2. Related Work

### 2.1. Vision Language Pre-training

Vision-language pre-training (VLP) seeks to improve the performance of downstream tasks that involve both vision and language by training models on extensive datasets of image-text pairs. A prominent class of vision-language models leverages contrastive learning (Alayrac et al., 2020; Cherti et al., 2023; Jia et al., 2021; Radford et al., 2021; Yao et al., 2021; Yuan et al., 2021; Zhai et al., 2023). Among them, CLIP (Radford et al., 2021), which was trained on a private WIT-400M dataset of image-text pairs, demonstrates impressive zero-shot capabilities across various downstream tasks, including image-text retrieval and image classification through text prompts. Furthermore, CLIP shows potential for application in broader fields such as decision making and robotics, where robots are required to perform language-

specified tasks based on visual inputs.

Recent multimodal foundation models, such Qwen-VL (Bai et al., 2023) and Llama 3-V (Dubey et al., 2024), all follow a similar pattern: they extract visual features by finetuning language-aligned vision models and train cross-attention layers to inject visual features into the language model. However, many researchers have observed that when data is scarce, fine-tuning or even applying additional layers on top of CLIP (instead of using raw CLIP features) (Kerr et al., 2023; Lan et al., 2024) may result in models with weaker reasoning capabilities compared to vanilla CLIP. This motivates our approach in OTTER, where we preserve CLIP's strong vision-language alignment capabilities by keeping the model frozen and extracts visual patch feature that correspond to the text query in a parameter-free way.

## 2.2. Vision Language Action Models

In recent years, there has been a surge of interest in developing robot foundation models, largely inspired by the success of large language models (LLMs) and vision-language models (VLMs) (Devlin et al., 2018; Radford et al., 2018; 2019; Brown et al., 2020; Chowdhery et al., 2023; Achiam et al., 2023; Radford et al., 2021; Li et al., 2023). A key hypothesis driving this trend is that more capable robot foundation models can emerge by scaling up robot datasets, increasing model capacity, and co-training or pre-training models on vision and language datasets. This has led researchers in the robot learning community to train robot foundation models, investigate pre-training strategies, and iterate on model designs (Brohan et al., 2022; 2023; Kim et al., 2024; Octo Model Team et al., 2024; Jang et al., 2022; Jiang et al., 2023; Reed et al., 2022; Collaboration et al., 2024; Shah et al., 2023; Fu et al., 2024).

Many existing VLMs (Liu et al., 2023; Laurençon et al., 2024; Karamcheti et al., 2024) use a similar approach, where visual features and languages are directly passed into the LLM to generate answers. Similarly, the majority of Vision-Language-Action (VLA) models also opt for this approach, where language, vision, and robot proprioception data are separately encoded by modality-specific feature extractors before being fed into a single transformer policy. This method has shown promise in many language-conditioned multi-task learning models (Jiang et al., 2023; Brohan et al., 2023; Jang et al., 2022; Reed et al., 2022; Collaboration et al., 2024; Shah et al., 2023), including current open-source state-of-the-art models such as Octo (Octo Model Team et al., 2024) and OpenVLA (Kim et al., 2024).

In contrast to the above approach, OTTER combines vision and language input before feeding them into the robot policies by extracting and passing text-aware visual features. Early works such as FiLM (Perez et al., 2018) encode text information and fuse these features into each block of a

ResNet (He et al., 2016). RT-1 (Brohan et al., 2022), one of the first language-conditioned robot policies, uses FiLM to encode visual and text information for action generation. However, RT-1 learns the language-vision alignment from robotic data without leveraging pre-trained models such as CLIP (Radford et al., 2021), where visual features are already aligned with text.

OTTER distinguishes itself by retrieving CLIP's visual patch features that best correspond to the language task description using cosine similarity before the policy transformer. These text-aware visual features, language features and the robot's proprioceptive state are fed into the robot policy. This approach allows the model to leverage the fine-grained features from the pre-trained vision-language models while incorporating robot-specific information.

## 3. Method

We propose OTTER, a vision-language-action model for learning a robot manipulation policy through extraction of text-aware vision features from a pre-trained VLM. We first describe how OTTER utilizes the vision-language alignment of pre-trained vision and language encoders to extract text-aware vision features, then provide a more detailed explanation of the model architecture.

### 3.1. Text-Aware Visual Feature Extraction

OTTER utilizes a pre-trained CLIP for vision and language features extraction. Consider a ViT-based CLIP vision encoder (Radford et al., 2021) consisting of a series of residual attention blocks. Each of these blocks takes as input a collection of visual tokens $X = [x_{\text{cls}}, x_1, \ldots, x_{h \times w}]^T$, where $x_{\text{cls}}$ represents the learnable global class token, and outputs the feature $X_{out}$ as shown below:

$$q, k, v = \text{Proj}_{q,k,v}(\text{LN}(X)) \tag{1}$$

$$X_{\text{sum}} = X + X_{\text{attn}} = X + \text{Proj}(\text{Attn}(q, k, v)) \tag{2}$$

$$X_{\text{out}} = X_{\text{sum}} + \text{FFN}(\text{LN}(X_{\text{sum}})) \tag{3}$$

Proj, LN, and FFN denote linear projection matrix, layer norm (Ba, 2016), and feed-forward network respectively. A recent work ClearCLIP (Lan et al., 2024) demonstrates that CLIP's last self-attention block's attention feature $X_{\text{attn}}$ contains cleaner semantic information than the CLIP's output feature $X_{\text{out}}$. While ClearCLIP uses the cosine similarity between text and visual features for segmentation, we leverage this similarity to construct task-relevant visual features for robotic control. Specifically, we use the similarity scores to select and combine visual features that best align with the task instruction, creating compact representations for downstream action prediction.

In OTTER, we extract text per-token features from CLIP's language encoder $f_l$ ($m$ tokens). For the visual features,

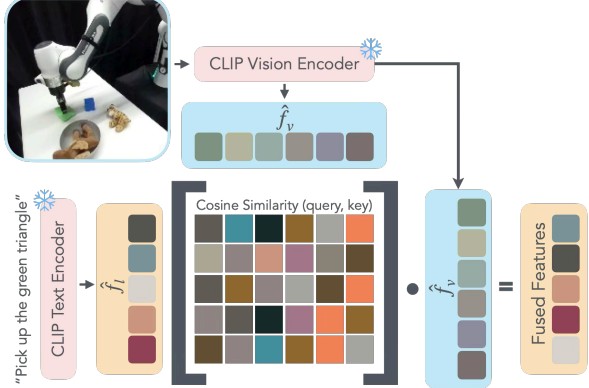

Parameter-Free Vision-Language Fusion

**Figure 3: Text-aware Visual Features Extraction** We calculate the similarity between the visual patch features and per-token language features, then take the softmax over the patch feature dimension. Intuitively, this gives a distribution of semantic similarity over all spatial locations. We then multiply the visual patch features to retrieve the visual semantic features that correspond to each token in the sentence. In Appendix C, we provide more in-depth analysis and visualizations of the proposed method.

motivated by the improved ability of ClearCLIP to capture text-aligned visual features, we specifically utilize the attention output $X_{\text{attn}}$ from the last vision attention layer, rather than the CLIP's output feature $X_{\text{out}}$, denoting it as $f_v$ (n tokens), where $n = h \times w$ is the total number of patch tokens from ViT. Figure 6 demonstrates how using $X_{\text{attn}}$ enhances the alignment between visual features and language semantics, illustrating the effectiveness of this approach. More details are in Appendix C.

Since the language features and the visual features have different dimensions, CLIP uses a matrix per modality to project the network's output feature to the same latent dimension, denoted as $w_l$ and $w_v$ for language and vision respectively. We normalize the text and visual features for vision-language fusion. The text features are normalized using the final layer normalization: $\hat{f}_l = \text{LN}_{\text{final}}(f_l)w_l$. The visual features are normalized using the post-attention layer normalization: $\hat{f}_v = \text{LN}_{\text{post}}(f_v)w_v$. We apply L2 normalization to both text and visual features: $\hat{f}_l = \hat{f}_l/\|\hat{f}_l\|_2$ and $\hat{f}_v = \hat{f}_v/\|\hat{f}_v\|_2$ as in standard CLIP.

With the normalized features, we perform temperature-weighted attention:

$$f_{vl} = \text{softmax}(\hat{f}_l \hat{f}_v^\top / \tau)(\hat{f}_v + PE) \quad (4)$$

where $\tau$ is the temperature parameter. $PE$ is the 2D sin-cos position embedding of the patch (Dosovitskiy et al., 2020), which informs the policy network of the spatial location of each patch. Same as in CLIP (Radford et al., 2021), $\tau$ is learnable and is clipped between 0 and 100. The resulting feature $f_{vl} \in \mathbb{R}^{m \times d}$ are the text-aware visual tokens, where each row is a linear combination of normalized visual

features $\hat{f}_v$. Intuitively, the *softmax* serves as a selection function, where patch features relevant to a particular language token are selected, and a weighted average of these patches is calculated to provide cues to where the robot policy should pay attention to. A smaller $\tau$ sharpens the *softmax*, concentrating the selection on the patch with the most similar feature, while a larger $\tau$ produces a smoother, more evenly distributed selection across patches. Critically, only $\tau$ is learnable and the entire CLIP model is *frozen* throughout training.

### 3.2. Model Architecture

**Policy Network Input** We compress the extracted text-aware visual features $f_{vl}$ into a single token for each camera. To achieve this, we apply a *learnable* cross-attention pooling operation to each camera's $f_{vl}$ to obtain a single feature $f'_{vl}$. Specifically, we use $N_q = 4$ learnable queries $q$, and keys $k$ and values $v$ from $f_{vl}$, and compute the output using cross attention $X_{attn}(q, k, v)$. We concatenate the $N_q$ output tokens to one single token ($f'_{vl}$). To facilitate better instruction following capabilities, we additionally employ another *learnable* cross-attention pooling on the text features $f_l$, resulting in a single text token $f'_l \in \mathbb{R}^{d_l}$. The robot's proprioceptive state is encoded through a Feed-Forward Network (FFN) to extract an embodiment feature $f_e$. At time step $t$, we concatenate the embodiment feature $f_e$ with the perception feature $f'_l$ and $f'_{vl}$ along the channel dimension to create a single token $f_t$. This token serves as input to a policy network for action prediction.

**Policy Network and Action Head** OTTER uses a transformer as the policy network, consisting of 4 layers and 8 heads, with a hidden dimension of 512. Fed by the combined features from the perception and embodiment, the model generates an action $a_t$. The model is trained with a context length of $T = 12$ steps. For each output token at a given timestep, we use a FFN to predict the next 12 actions. More details about our model architecture can be found in Appendix B.

**Proprioception Parametrization** We parameterize the proprioception space using a 10-dimensional representation. This includes the absolute end effector translation (x, y, z), a 6DoF rotation vector, and a continuous end-effector gripper state. The 6DoF rotation vector is derived by flattening the first two rows of the $SO(3)$ rotation matrix.

**Action Parametrization** We employ delta end effector pose as our action parameterization. At each prediction step, the model predicts $t$ actions. Given a sequence of *absolute* end effector action transforms $T_1, T_2, \cdots, T_t$ in a trajectory and the current end-effector pose $T_{\text{ee}}$, we define the relative transforms that the model needs to predict as $T_{\text{ee}}^{-1}T_1, T_{\text{ee}}^{-1}T_2, \cdots, T_{\text{ee}}^{-1}T_t$. We then append the continuous absolute gripper position to each delta action. Similar

Simulation Scenes          Physical Scenes

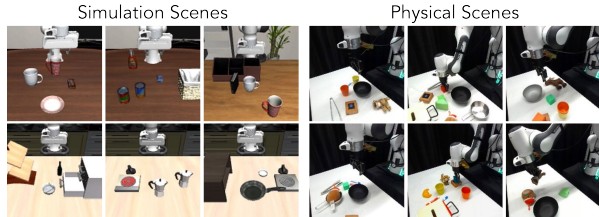

**Figure 4:** Example scenes in the simulation (left) and in the physical environments (right) using a Franka robot.

to the proprioception representation, we express the delta action using the relative end effector translation and a 6DoF rotation vector, resulting in a 10-dimensional action representation. When executing the predicted actions, we employ temporal ensembling (Zhao et al., 2023) in conjunction with receding horizon control (Chi et al., 2023). Through experimentation, we determined that an action horizon of 8 steps yields optimal performance.

## 4. Experiments

We consider two classes of problems: language-conditioned multi-task learning and zero-shot generalization in unseen environments. For language-conditioned multi-task learning, given a multi-task setup (defined as in there are many tasks that can be performed in the same scene), the policy needs to perform the correct task corresponding to the language instruction. In the zero-shot generalization setup, the policy is provided with a language description of an unseen task, and is asked to perform the specified task in the unseen environments. We introduce our experimental setup to evaluate the instruction-following and text-aware visual features extraction generalization of OTTER in Section 4.1 and the baselines considered in this paper in Section 4.2.

### 4.1. Environment Setup

**Simulation Environment** We use the LIBERO benchmark (Liu et al., 2024) for simulation evaluation. Specifically, we use the tasks and datasets in LIBERO-Spatial, LIBERO-Object, LIBERO-Goal, and LIBERO-90, which contains diverse objects, scene layouts, and language instructions. Each simulation task has 50 demonstrations. We evaluate OTTER's capabilities on both in-distribution tasks and unseen tasks. The in-distribution tasks are the 30 tasks in the original LIBERO-Spatial/Object/Goal, which can evaluate the model's multi-task learning capabilities. In addition, we also construct 10 novel tasks, where we modify the language instructions and corresponding objects of 10 original tasks from LIBERO-90. For the 10 unseen tasks, we follow the same convention in LIBERO (Liu et al., 2024) about object initialization and goal configuration by defining task bddl files. Example scenes in the simulation are shown in the left column of Figure 4.

**Real Robot Environment** For real-robot evaluation, we

consider four robotic task primitives: pick up and place, poking, pouring and opening/closing a drawer. We define task as the combination of the primitive and objects involved. We collect robotic datasets on multi-task scenes using a Franka robot, where there are multiple tasks that can be completed in the same scene. We consider 10 pick-and-place tasks each containing 50-80 demonstrations of human tele-operating the robot, resulting a total of 724 demonstrations. We denote this dataset as DS-PnP. For each of the other three primitives we collect 100-200 demonstrations, with a total of 1,185 demonstrations. We denote the dataset consisting of all four primitives as DS-ALL.

We consider a task with *unseen* target objects for task completion as an unseen task. The training tasks involve objects encountered during model training, whereas the unseen tasks test the model's ability to generalize to unseen objects or scenes. For example, the model is trained on the task of poking a wooden block and is tested on the new task of poking a radish. Example scenes in real are shown in the right column of Figure 4.

We consider 19 in-distribution training tasks and 15 out-of-distribution unseen tasks across the 4 primitives (Table 6). For each task, we evaluate the model for 10 experiment trials. For each experiment trial, we vary the location of the target object and introduce 2-3 random distractor objects, to evaluate the instruction following capability of the VLA models. In the unseen tasks, we provide the robot with novel target objects that are unseen during training, or novel combinations of target objects and target placement locations. This setup aims to evaluate both object identification and task completion ability under more challenging and previously unseen conditions. The robot must identify and interact with the correct object based on the provided language instruction and complete the assigned task.

The trial is terminated either when the task is completed or when a time limit is reached. The overall performance is measured by calculating the average success rate with standard error across all trials for the training and unseen tasks. The full lists of simulation and real-world environments and more experiment details can be found in Appendix A.

### 4.2. Baselines

To evaluate if the text-aware visual features extracted in OTTER can better leverage the semantic understanding capabilities of the pre-trained VLMs, we consider four baselines that directly take all visual and language features, including three state-of-the-art open-sourced VLA models and one variant of OTTER:

1. Octo (Octo Model Team et al., 2024), an open-sourced transformer-based policy trained from scratch on 800K trajectories from the Open X-Embodiment dataset (Col-

| Method | Training Tasks | Unseen Tasks |
|---|---|---|
| $\pi_0$-Fast-Droid | - | $61\% \pm 5.3\%$ |
| Finetuned Octo | $15\% \pm 3.4\%$ | $12\% \pm 3.6\%$ |
| OTTER w.o. CLIP vision | $17\% \pm 2.9\%$ | $11\% \pm 2.5\%$ |
| Finetuned OpenVLA | $30\% \pm 3.9\%$ | $9\% \pm 3.1\%$ |
| DFP-OTTER | $29\% \pm 3.7\%$ | $4\% \pm 1.6\%$ |
| OTTER (Finetune CLIP) | $26\% \pm 4.0\%$ | $15\% \pm 3.9\%$ |
| OTTER w.o. $f_e$ | $40\% \pm 4.0\%$ | $29\% \pm 4.3\%$ |
| OTTER w.o. $f_l'$ | $57\% \pm 4.4\%$ | $53\% \pm 4.6\%$ |
| OTTER (Ours) | $68\% \pm 4.3\%$ | $62\% \pm 4.2\%$ |
| OTTER-OXE (Ours) | $\mathbf{72\%} \pm \mathbf{3.9\%}$ | $\mathbf{73\%} \pm \mathbf{2.8\%}$ |

**Table 1: Physical Single Primitive Multi-task Experiments.** For each model, we conduct physical robot pick and place experiments, with 100 trials on in-distribution training tasks and 70 trials on unseen tasks. OTTER achieves a similar success rate on the in-distribution training tasks and unseen tasks, significantly outperforming the baselines, highlighting the benefits of extracting text-aware visual features and a frozen pre-trained VLM.

laboration et al., 2024).

2. OpenVLA (Kim et al., 2024), a fine-tuned Prismatic-7B (Karamcheti et al., 2024) VLM on the Open X-Embodiment (OXE) dataset.

3. $\pi_0$-Fast-Droid (Black et al., 2024), a VLA using a pre-trained PaliGemma (Beyer et al., 2024) and a Fast (Pertsch et al., 2025) action tokenizer. This model is pre-trained on a subset of OXE and the $\pi$ dataset, and is fine-tuned on the Droid (Khazatsky et al., 2024) dataset.

4. Direct Feature Passing OTTER (DFP-OTTER): a variant of OTTER where the text tokens, vision tokens are passed to an attention pooling layer separately to obtain independent tokens, which are then concatenated with the embodiment feature $f_e$ as the input to the transformer. This baseline is constructed to inform the importance of text-aware visual feature extraction.

## 5. Results

We compare OTTER against several baseline in both real-world (Section 5.1) and simulation environments (Section 5.2). We compare OTTER against several ablations in Section 5.3. In Section 5.4, we further investigate OTTER's capabilities by scaling up models.

### 5.1. Real-world Experiments

**Single Primitive** We first evaluate all models on both the training and unseen tasks of the pick and place primitive in the real robot environment, as shown in Table 1. For fair comparisons, we fine-tune Octo and OpenVLA on DS-PnP using the same amount of learning steps. We compare the performance of OTTER trained from scratch and OTTER-OXE pre-trained on the OXE dataset and fine-tuned on

DS-PnP. For $\pi_0$-Fast-Droid model, since our experiment setup is the Droid setup, we directly evaluate this model on our setup, denoted as $\pi_0$-Fast-Droid. More details about model training and architectures are in Appendix B.

For unseen pick up and place tasks, $\pi_0$-Fast-Droid is able to achieve a success rate of 61%. This is because this model is fine-tuned on Droid, which contains many pick up and place tasks. In both the training and unseen tasks, Octo struggles to accurately identify the object of interest and determine the correct placement location, leading to a low success rate. We hypothesize this can be attributed to two key factors. First, Octo does not incorporate a pre-trained VLM, such as CLIP, into its network. Instead, it trains its vision encoder from scratch using a large-scale robotic dataset (OXE (Collaboration et al., 2024)), which lacks the semantic diversity found in larger vision datasets like LAION (Schuhmann et al., 2022). Second, OTTER extract text-aware visual features from the pre-trained CLIP, which results in a strong vision language association. This enables better visual grounding and generalization capabilities of OTTER to perform better on training and unseen tasks, despite being trained on a small robotic dataset. OpenVLA and DFP-OTTER perform similarly, which is better than Octo on training tasks, but much worse than OTTER. On unseen tasks, they both fail to generalize. We hypothesize this is because it's challenging for the direct feature passing architectures to learn generalizable vision-language connections on a small robotic dataset, while OTTER can utilize the extracted text-aware vision features from the pre-trained VLM. OTTER-OXE performs better than OTTER on both training and unseen tasks, suggesting that OTTER's performance scales with more data, possibly because the policy network can learn better and more precise action planning from more robotics data.

**Multiple Primitives** We evaluate all models on all four primitives. We compare the performance of Octo and OpenVLA finetuned on DS-ALL and OTTER pretrained on OXE and fine-tuned on DS-ALL, denoted as OTTER-OXE. As there are more primitives, we also consider a deeper and wider OTTER model (details in Table 7), denoted as OTTER-L. For a fair comparison, we extended the context history length of Octo to 10 (Octo cannot exceed a context length of 10 due to its inherent design constraints) and matched its action prediction horizon to ours. As OpenVLA has many tokens per timestep, its context length cannot be extended and we use its default context length. For $\pi_0$-Fast-Droid model, as in the single primitive experiment, we directly evaluate this model on our setup. As most of the Droid tasks are pick and place while our setup contains other primitives, we also consider fine-tuning $\pi_0$-Fast-Droid on the DS-ALL, denoted as Finetuned $\pi_0$-Fast-Droid.

Results are shown in Table 2. All models are evaluated

| Method | Pouring | Drawer | Poking | Pick and Place | Mean±Std. Err. |
|---|---|---|---|---|---|
| $\pi_0$-Fast-Droid | 0% | 0% | 0% | 61% | 29% ± 3.5% |
| Finetuned $\pi_0$-Fast-Droid | 0% | 45% | 27% | 51% | 35% ± 3.8% |
| Finetuned Octo | 0% | 0% | 0% | 5% | 4% ± 1.2% |
| Finetuned OpenVLA | 0% | 0% | 0% | 1% | 0.6% ± 0.5% |
| OTTER | 63% | 50% | **93%** | 61% | 67% ± 3.8% |
| OTTER-L | 65% | 55% | 90% | 69% | 71% ± 3.5% |
| OTTER-OXE | 60% | 65% | **93%** | 66% | 70% ± 3.6% |
| OTTER-OXE-L | **77%** | **75%** | **93%** | **75%** | **77% ± 3.3%** |

**Table 2: Multi-primitive zero-shot generalization**: We train models across four manipulation primitives (pouring, drawer manipulation, poking, and pick-and-place) with a total of 1,185 human tele-operated demonstration trajectories and evaluate them on 150 trials of **completely unseen tasks** within these primitives. Despite the inherent difficulty of zero-shot generalization across multiple primitives, OTTER achieves significantly higher success rates compared to baselines across all primitives. While baseline models struggle with generalization, particularly on pouring task (0% success rate), OTTER maintains substantial performance (60-93% success rate) on unseen tasks. The results further suggest that OTTER's generalization capabilities can be enhanced through increased model capacity (OTTER-L) and pre-training on large robotic datasets (OTTER-OXE).

on **unseen** tasks for each primitive, with 10 trials for each task. While $\pi_0$-Fast-Droid achieves decent performance on the pick and place primitives, it fails on all the other three primitives as the majority of the Droid dataset is the pick and place primitive. Finetuned $\pi_0$-Fast-Droid achieves non-zero success rate on Drawer and Poking primitives, but still fails on the pouring primitive. There is also a degraded performance on the Pick and Place primitives for Finetuned $\pi_0$-Fast-Droid. This highlights the difficulty of our experiment setup. The performance of Octo, Open-VLA, and OTTER-OXE on the pick and place task all drop, showing the difficulty of multi-primitive learning. Notably, both Octo and OpenVLA fail to complete any unseen tasks for the pouring, drawer, and poking tasks, likely due to a relatively small amount of demonstrations provided for each primitive. Both OTTER-OXE and OTTER-OXE-L can achieve high success rate on all four primitives on the same amount of demonstrations, indicating that using text-aware visual features extracted from a pre-trained VLM can increase the data efficiency and enhance the generalization ability. OTTER-L outperforms OTTER on average, with the performance gap increases as OTTER pre-trained on the OXE dataset, indicating that OTTER can scale with model size.

### 5.2. Simulation Experiments

We compare OTTER with various baselines in simulation, as shown in Table 3. For fair comparison with OpenVLA and Octo, we process the simulation datasets following the procedure described in the OpenVLA paper (Kim et al., 2024), and use their reported performance on the standard LIBERO tasks. For the unseen tasks, we change 10 tasks in LIBERO-90 with different objects and distractors to test the generalizability of all the models on unseen tasks. From Table 3, we found all the models perform similarly on training tasks in LIBERO due to the limited variations of the tasks

and sufficient demonstration datasets. For tasks in LIBERO-Object and LIBERO-Spatial, DFP-OTTER is worse than OTTER because OTTER does better in localizing the object to interact with. Note that Octo and OpenVLA are pre-trained on the OXE dataset but DFP-OTTER is trained from-scratch on LIBERO, so DFP-OTTER's performance is worse. In unseen tasks, OTTER can outperform other baselines by a large margin, demonstrating its generalization capabilities to novel scenarios, which is consistent with the conclusion of the real-world experiments.

### 5.3. Ablations

We perform ablation studies with single-primitive, multi-task setting, where all models are trained on DS-PnP. We consider the following ablations on the design choices of OTTER. Addition ablation studies can be found in Appendix D.

1. OTTER w.o. $f_e$: OTTER without the embodiment representation $f_e$. The concatenated text token $f_l'$ and fused vision-language token $f_{lv}'$ are passed as the input to the transformer.

2. OTTER w.o. $f_l'$: OTTER without the text token $f_l'$. Only $f_{lv}'$ and $f_e$ are concatenated as the input to the transformer.

3. OTTER w.o. CLIP vision: OTTER using a smaller ViT to train from scratch instead of a frozen pre-trained CLIP vision encoder.

4. OTTER (Finetune CLIP): OTTER with the CLIP initialized from the pre-trained weight and fine-tuned end to end on the robotic dataset.

**Real-world Results** Physical results in Table 1 suggest that the performance on both the training tasks and the unseen

| Method | LIBERO-Spatial | LIBERO-Object | LIBERO-Goal | Train (Average) | Unseen |
|---|---|---|---|---|---|
| Finetuned Octo | 79% ± 1.0%* | 86% ± 0.9%* | **85% ± 0.9%*** | 83% ± 1.0% | 26% ± 1.1% |
| Finetuned OpenVLA | **85% ± 0.9%*** | 88% ± 0.8%* | 79% ± 1.0%* | 84% ± 0.9% | 48% ± 1.0% |
| DFP-OTTER | 79% ± 0.9% | 80% ± 1.0% | 78% ± 1.1% | 79% ± 1.0% | 45% ± 1.0% |
| OTTER (Ours) | 84% ± 1.0% | **89% ± 1.2%** | 79% ± 0.9% | 84% ± 1.1% | **61% ± 1.1%** |

**Table 3: Simulation results on LIBERO**. We evaluate OTTER and other baselines on 30 in-distribution tasks in LIBERO-Spatial/Object/Goal and on 10 unseen tasks we constructed, each task 50 trials. The numbers marked with ∗ of are directly referred from the OpenVLA (Kim et al., 2024) paper. The detailed training and evaluation on the unseen tasks are described in Section 4.1.

tasks drop significantly for the ablations compared to OTTER. The performance of OTTER without the embodiment features ($f_e$) drops 28% on the training tasks and 33% on the unseen tasks, indicating that $f_e$ is vital for task completion and generalization, likely because it provides a physical grounding for decision-making. Without $f_e$, the model's understanding of embodied features, possibly linked to the spatial or physical aspects of the task, is severely impaired. OTTER without the language features ($f_l'$) experiences a performance drop of around 10% on both training and unseen tasks, suggesting that $f_l'$ provides complementary information that may help in more nuanced task understanding.

OTTER w.o. CLIP vision has a significant performance drop of more than 50% on the training and unseen tasks in physical experiments. The results of OTTER w.o. CLIP vision is similar to Octo which also trains a vision encoder from scratch on the robotics dataset. This suggests that pretrained VLM provides more robust and transferable visual representations. Training a vision encoder from scratch can result in poor performance, as it lacks the generalization capabilities learned from large-scale pre-training.

OpenVLA demonstrates that fine-tuning the vision encoder of the pre-trained VLM on the robotics dataset is crucial for improving the performance. However, we found that fine-tuning the pre-trained vision encoder actually degrades the model's vision-language understanding capabilities. The large performance discrepancy between training and unseen tasks in both OpenVLA and OTTER (Finetune CLIP) suggests that fine-tuning compromises generalization ability, highlighting the benefits of OTTER's approach of extracting text-aware visual features from a frozen pre-trained VLM. We want to emphasize that both the effective feature extraction and the frozen encoder are crucial for learning a generalizable VLA, as shown by the worse performance of DFP-OTTER with a frozen VLM, OTTER (Finetune CLIP) that has a fine-tuned VLM and OpenVLA that is a VLA with direct feature passing and fine-tuned VLM.

**Simulation Results** Table 4 presents the simulation results of OTTER and other ablations. On the in-distribution tasks, OTTER w.o. CLIP vision and OTTER can work similarly well given sufficient demonstrations, but OTTER w.o. CLIP vision is more than 50% worse on unseen tasks, which shows the benefits of using a pre-trained VLM for better generalization capabilities.

| Method | LIBERO-Object | Unseen |
|---|---|---|
| OTTER w.o. CLIP vision | 80% ± 0.7% | 29% ± 0.9% |
| OTTER w.o. $f_e$ | 79% ± 0.9% | 48% ± 0.8% |
| OTTER w.o. $f_l'$ | 71% ± 1.2% | 49% ± 1.0% |
| OTTER (Ours) | **89% ± 1.2%** | **61% ± 1.1%** |

**Table 4:** Ablation results on LIBERO Object tasks and unseen tasks. We evaluate OTTER and other baselines on 100 trials of in-distribution LIBERO-Object tasks, and 100 trials of unseen tasks.

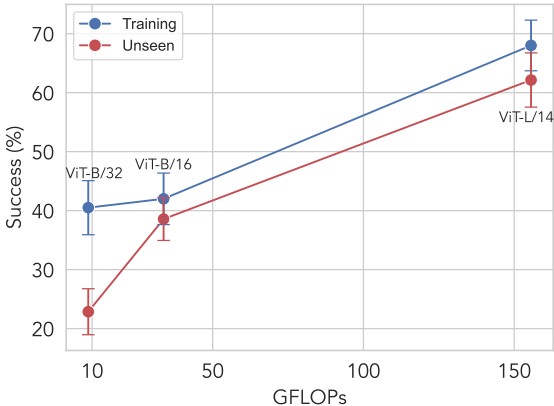

**Figure 5:** We evaluate OTTER's performance with improved vision language features by scaling CLIP. In particular, we train OTTER with three CLIP variants with increasing FLOPs: ViT-B/32, ViT-B/16, and ViT-L/16. We report the task performance vs. the inference FLOPs per image on training and unseen tasks. The results suggest that the OTTER can benefit from scaling up vision-language model.

The performance of OTTER w.o. $f_e$ drops about 10% on both the in-distribution and unseen tasks, indicating that $f_e$ is beneficial for task completion as it provides explicit spatial information of the robot. OTTER w.o. $f_l'$ is also noticeably worse. We hypothesize this is due to the object are not very realistic in simulation, so the extracted text-aware visual features in CLIP may highlight multiple objects or wrong objects. $f_l'$ can provide complementary information for the policy to interact with the correct objects.

## 5.4. Scaling up Vision and Language Encoders

Prior work has shown that vision-language encoders exhibit improved semantic understanding capabilities as their computational complexity increases (Radford et al., 2021). To investigate whether OTTER can leverage these improvements, we evaluated its performance using three CLIP model vari-

ants of increasing computational complexity: ViT-B/32, ViT-B/16, and ViT-L/14. As shown in Figure 5, OTTER's task success rate improves substantially as we scale up the CLIP model's FLOPs, with gains observed on both training (+27.5%) and unseen pick-and-place tasks (+39.3%) when moving from ViT-B/32 to ViT-L/14. These results demonstrate that OTTER effectively utilizes larger vision-language encoders for enhanced semantic understanding and serves as a scalable method for integrating large-scale vision-language pre-training with learning from robotic datasets.

## 6. Limitations and Conclusions

While OTTER demonstrates improved task completion rates compared to existing VLAs, it still faces several limitations. One significant challenge is scaling across different morphologies, particularly those that cannot be easily parameterized by SE(3) transforms (i.e. robot multi-finger hand). This limitation restricts the model's adaptability to a wider range of robotic platforms and task types. Furthermore, this study has not extensively explored how this method scales to long-horizon tasks and more complex scenes, which could be an important area for future research.

In summary, we present OTTER, a vision-language-action model that leverages text-aware visual feature extraction from pre-trained vision-language encoders. By utilizing the semantic alignments during large-scale vision-language pre-training, OTTER achieves significantly better generalization than existing VLA models across robot manipulation tasks sampled from multiple motion primitives, maintaining higher success rates on *unseen objects and environments* where previous methods fail. The experiments demonstrate that OTTER's performance scales along multiple axes: through larger pre-trained vision-language encoders, increased policy network capacity, and pre-training on larger robot datasets. These results suggest that by better preserving existing alignment in pre-trained vision-language encoders, rather than learning them directly from robotics data, is beneficial for developing more capable and generalizable robot learning systems.

## Acknowledgement

Dataset usage and model training work solely conducted at Berkeley. Pieter Abbeel holds concurrent appointments as a Professor at UC Berkeley and as an Amazon Scholar. This paper describes work performed at UC Berkeley and is not associated with Amazon. This research was performed at the AUTOLAB at UC Berkeley in affiliation with the Berkeley AI Research (BAIR) Lab, and the CITRIS "People and Robots" (CPAR) Initiative. In their academic roles at UC Berkeley, Letian Fu, Fangchen Liu, Huang Huang, and Ken Goldberg are supported in part by donations from Meta, Autodesk, Google, Siemens, Toyota Research Institute, Bosch, and by equipment grants from PhotoNeo, Nvidia, NSF AI4OPT Centre, and Intuitive Surgical.

## Impact Statement

This paper presents work whose goal is to advance the field of machine learning and robotics. There are many potential societal consequences of our work, none of which we feel must be specifically highlighted here.

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

# A. Environment Setup

## A.1. Simulation Tasks

For the training tasks, we use the original tasks in LIBERO-Goal, LIBERO-Spatial, and LIBERO-Object. We also build unseen evaluation tasks based on 10 original LIBERO-90 tasks, by changing language instructions and target object color and type in the task bddl files. The 10 unseen tasks are listed in Table 5.

| Changes | Unseen |
|---|---|
| object type | Put the moka pot in the bottom drawer of the cabinet |
| object type | Put the moka pot on the wine rack |
| object type | Pick up the ketchup and put it in the basket |
| object type | Pick up the ketchup on the plate |
| object type | Pick up the bottle and put it in the tray |
| object color | Put the black bowl on top of the cabinet |
| object color | Put the black bowl on the plate |
| object color | Put the red mug to the right of the plate |
| object color | Put the yellow and white mug in the front of the red mug |
| object color | Put the red mug to the front of the moka |

**Table 5:** The 10 in-distribution tasks and 7 unseen tasks we used in our real-world setting.

## A.2. Real-world Tasks

The full list of tasks for our real-world evaluation is provided in Table 6.

| In-Distribution | Unseen |
|---|---|
| Put potato in pot to black bowl | Put yellow cube in black bowl |
| Pickup potato | Pick up radish and place it in grey bowl |
| Pick up and place deer in grey bowl | Put blue bear in pink bowl |
| Pick up green triangle | Put yellow cube in grey bowl |
| Put tiger to black bowl | Put apple with a green leaf in black bowl |
| Put red cube into black bowl | Pick up blue sponge and place it in steel pot |
| Put blue cube into grey bowl | Pick up black dog and place it in the pink bowl |
| Put the red ball in black bowl | |
| Put green triangle into pink bowl | |
| Put blue cube in pink bowl | |
| Poke a wooden block | Poke the radish |
| Poke a tiger | Poke the gray dog |
| Poke a green triangle | Poke the pink bowl |
| Poke a gray bowl | |
| Pour from the brown cup to the gray bowl | Pour from the orange cup to the black bowl |
| Pour from the blue cup to the pink bowl | Pour from the blue cup to the black bowl |
| Pour from the yellow cup to the black bowl | Pour from the brown cup to the pink bowl |
| Open the drawer | Open the drawer with a tiger on top |
| Close the drawer | Close the drawer with a red cube inside |

**Table 6:** The 10 in-distribution tasks and 7 unseen tasks we used in our real-world setting.

For each experiment trial of poking and pouring, we vary the location of the target object to manipulate and introduce 2 or 3 random distractor objects. For drawer, we vary the location of the drawer on each trial. Similar to the pick and place primitive, for each task, we generate 10 randomized scenes.

Each trial is scored based on the robot's performance in completing the task. For the pick and place primitive, a score of **0.5** is awarded if the robot successfully picks up the correct target object, and a score of **1** is given if the robot not only picks up the correct object but also places it in the correct location as specified by the instruction. If the robot fails to pick up the target object or picks up a distractor object, a score of **0** is recorded. For other primitives, a score of **1** is recorded if the task is completed, otherwise a score of **0** is given.

For all models other than the OpenVLA, each trial is allowed a maximum of 30 seconds to complete. As OpenVLA is a large 7B model wit a lower inference speed, we give it a time limit of 60 seconds to complete a task.

## B. Model and Training Details

### B.1. Model Architecture for OTTER and Baselines

The details of our model parameters can be found in Table 7. All the baselines share the same hyper-parameters with OTTER. For OTTER w.o. CLIP Vision, we use a ViT Encoder based on the implementation of `https://github.com/google-research/vision_transformer` with a ViT-Ti/16 configuration with half of the number of attention layers. For OTTER w.o. $f_e$ and OTTER w.o. $f_l'$, we use the same model configuration but only remove the corresponding attention pooling layers. We incorporate action chunking into OpenVLA by asking it to predict the next 16 actions, which performs better than vanilla OpenVLA which predicts only the next step. For Octo, we use the official Hugging Face Checkpoint at `hf://rail-berkeley/octo-small-1.5` which is in a comparable size with our model. During inference, we cache the CLIP feature outputs. This enables the ViT-L/14 OTTER model to perform inference at $50Hz$ on a single NVIDIA 3090Ti, allowing real-time control.

| Hyperparameter | Value |
|---|---|
| CLIP Model (Real) | ViT-L/14 |
| CLIP Model (Sim) | ViT-L/32 |
| # Pooling Readouts | 4 |
| # Pooling Attention Heads | 8 |
| # Pooling Attention Blocks | 2 |
| # Text-Pooling Output Dimension | 128 |
| # Image-Pooling Output Dimension | 512 |
| # Proprio-Pooling Output Dimension | 64 |
| *Causal Transformer Parameters:* | |
| # Attention Blocks | 4 (8) |
| # Attention Heads | 8 |
| # Latent Dimension | 512 (768) |
| # Context Length (Real) | 12 |
| # Context Length (Sim) | 4 |
| # Action Prediction Horizon (Real) | 12 |
| # Action Prediction Horizon (Sim) | 1 |
| # Parameter | 11,790,522 (25,516,986) |

**Table 7:** Hyperparameters for OTTER model architecture. Values in the parenthesis shows the hyperparameters for a larger and wider OTTER for the real world experiments.

### B.2. Training Hyper-parameters

We use the AdamW optimizer with a cosine learning rate decay schedule and linear learning rate warm-up. We list training hyperparameters in Table 8. All these hyper-parameters are shared between real-world and simulation. All the models are trained on 4 NVIDIA A100 80GB GPUs.

## C. Vision-Language Attention Visualization

To provide further motivations for why using $X_{out}$ (per (Lan et al., 2024)) instead of the output feature map of CLIP, we compare the cosine similarity between the output of the CLIP ViT-L/16 encoder and the per-token text features in three different settings: (1) fine-tuning the encoder, (2) a frozen CLIP's output features ($X_{out}$), and (3) a frozen CLIP's last attention block's feature ($X_{attn}$) as described in Section 3.1. The similarity visualization is shown in Figure 6.

It may initially seem unexpected that this type of visualization is reasonable. However, this can be explained by the fact that LayerNorm operates independently of the patch dimension, as it normalizes along the channel dimension. When combined with the vision-alignment weight matrix $w_v$, the operation $\hat{f}_v = \text{LN}_{\text{post}}(f_v)w_v$ remains linear. Therefore we can linearize the final attention block:

$$\hat{f}_v = \text{LN}_{\text{post}}(X_{out})w_v \tag{1}$$

$$= \text{LN}_{\text{post}}(X_{res} + X_{attn} + \text{FFN}(\text{LN}(X_{sum})))w_v \tag{2}$$

$$= \text{LN}_{\text{post}}(X_{res})w_v + \text{LN}_{\text{post}}(X_{attn})w_v + \text{LN}_{\text{post}}(\text{FFN}(\text{LN}(X_{sum})))w_v \tag{3}$$

| Hyperparameter | Value |
| --- | --- |
| Learning Rate | 3e-4 |
| Warmup Steps | 2000 |
| Weight Decay | 0.01 |
| Learning Rate Scheduler | cosine |
| Gradient Clip Threshold | 1 |
| Batch Size | 64 |
| Total Gradient Steps | 40000 (60000) |
| Image Resolution | $224 \times 224$ |
| Random Resized Ratio | [0.9, 1.1] |
| Random Brightness | 0.2 |
| Random Contrast | [0.8, 1.2] |
| Random Saturation | [0.8, 1.2] |
| Random Hue | 0.1 |

**Table 8:** Hyperparameters used for training (pre-training on OXE).

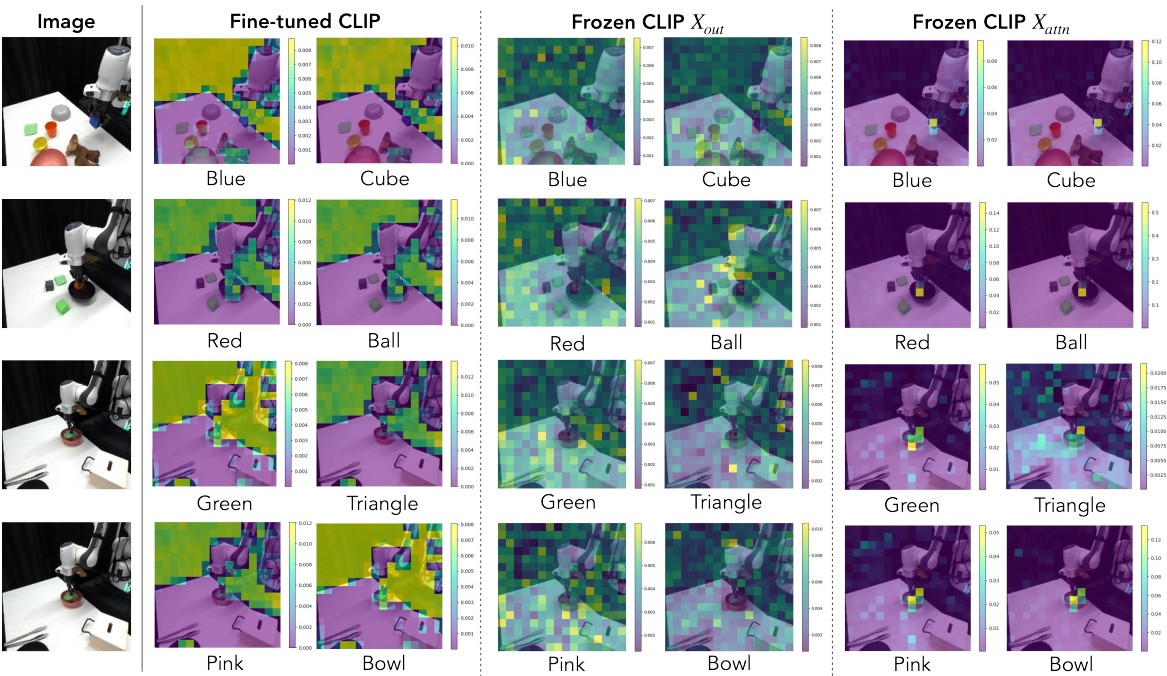

**Figure 6:** Examples of attention maps for CLIP fine-tuned with VLA (left) and frozen CLIP's output ($X_{out}$) (middle) and frozen CLIP's attention features ($X_{attn}$) (right). The first column shows the side view observation and the text query is below each attention map. Fine-tune CLIP pays attention to the background and the frozen CLIP's output ($X_{out}$) is noisy. In contrast, the frozen CLIP ($X_{attn}$) pays attention to the correct object associated with the text query. These examples indicate that fine-tuning CLIP on robotic datasets can degrade the performance of the pre-trained CLIP, especially when the robotics dataset is small. It also highlights the benefits of using $X_{attn}$ for fused vision-language features.

For ClearCLIP, or Frozen CLIP $X_{attn}$, we are visualizing the $\text{LN}_{\text{post}}(X_{attn})w_v$ term.

Similar to what ClearCLIP has noted, after adding residual connection and the final FFN, the features become noisy and worsen the alignment between language and visual features. The noisy attention map makes it challenging for the model to identify the correct features directly from the feature map, which makes it necessary for existing VLA (i.e. OpenVLA (OpenAI, 2024)) to fine-tune the CLIP vision encoder. In comparison, by using $X_{attn}$, object localization becomes an easier task in OTTER: we can extract the location of the object by getting the *softmax* across the attention map without using any parameters (see Figure 3). More attention map examples on Open-X dataset are in Figure 7.

In the finetuning v.s. frozen CLIP ($X_{attn}$) comparison, fine-tuning OTTER's CLIP results in overfitting to foreground-background separation, causing it to lose zero-shot object detection ability. This limits the model's ability to highlight the correct object, leading to a significant drop in task success rates (26% vs 68% for training tasks and 15 vs 62% for unseen tasks). Conversely, a frozen CLIP ($X_{attn}$) preserves object detection capabilities, providing better downstream performance.

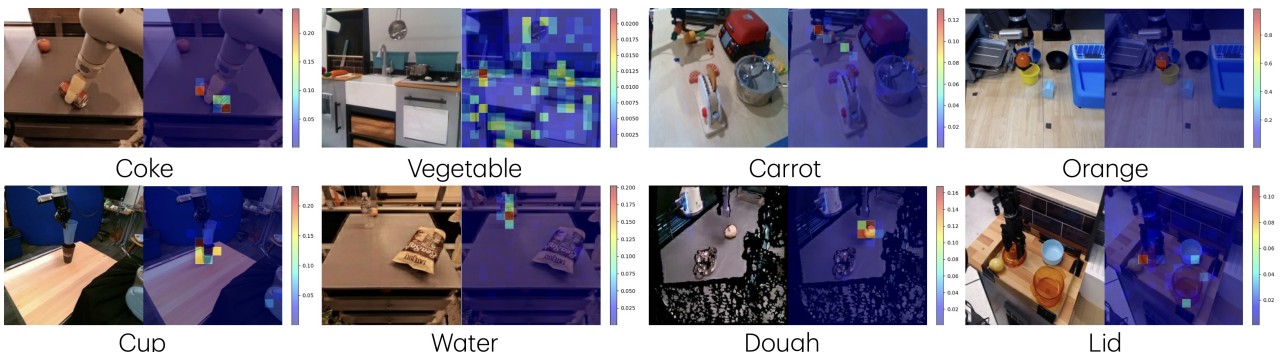

**Figure 7:** Examples of attention maps of frozen CLIP's attention features (Xattn) on Open-X dataset. The bottom texts are the corresponding text tokens.

## D. More Ablations

We consider another 2 ablations of OTTER. Both are trained on the DS-PnP.

1. DFP-OTTER (CLS): another variant of OTTER that utilizes CLIP's ¡cls¿ token rather than text-aware visual feature extraction.

2. OTTER (xattn): OTTER using standard cross attention pooling between the text tokens $f_l$ and the vision tokens $f_v$ to obtain the fused vision language features $f'_{lv}$ instead of text-aware visual feature extraction as in Eq.( 4).

From Table 9, both DFP-OTTER (cls) and OTTER (xattn) fail to generalize to unseen tasks, highlighting the benefits of using text-aware visual feature extraction to obtain task-related vision features as the fused vision language features.

| Method | DFP-OTTER (CLS) | OTTER (xattn) | OTTER |
|---|---|---|---|
| Success Rate | $6\% \pm 0.8\%$ | $2\% \pm 0.5\%$ | $62\% \pm 4.2\%$ |

**Table 9:** Physical results on 70 trials on unseen pick up and place tasks for other variants of OTTER.

