# OpenReview forum: "OTTER: A Vision-Language-Action Model with Text-Aware Visual Feature Extraction"
_ICML.cc/2025/Conference — ICML 2025 poster_

### Official Review · Reviewer_Gn1m · 2025-03-13

**Overall Recommendation:** 3

**Summary:**

The authors propose a novel Vision-Language-Action robotic policy called OTTER. Motivated by the computational burden that fine-tuning VLMs for robot policy use entails, as well as by observations that fine-tuning VLMS for action generalization risks weakening the pre-trained vision-language alignment, the authors propose freezing vision and language encoders, extracting task-relevant visual features guided by language instructions. They compare OTTER to previous methods Octo and OpenVLA through experiments on Libero and in a real world setting, and ablate several aspects of OTTER in the real world setting.

**Claims And Evidence:**

Yes.

**Essential References Not Discussed:**

N/A

**Experimental Designs Or Analyses:**

Yes.

**Methods And Evaluation Criteria:**

Yes.

**Other Comments Or Suggestions:**

N/A

**Other Strengths And Weaknesses:**

Strengths:

S1: Technical Novelty: The idea of reducing the number of visual tokens input to the model is important for computational efficiency, and OTTER’s use of existing information from CLIP’s features to extract the most important visual tokens is elegant.

S2: Real world experiments: The real world experiments are well-designed and thorough. It is often difficult to get robot policies to work in real-world settings, as shown by Rows 1-2 in Table 2, so OTTER’s strong performance is very promising.

Weaknesses:

W1: Generalization: OTTER vastly outperforms Octo and OpenVLA on the real world experiments but only performs on par with them on the synthetic Libero benchmark. My concern here is that OTTER is tailored to the experimental real world set up and will not see the same boosts in performance on other environments.

W2: Ablations: Several of OTTER’s design decisions are not ablated, namely: whether to use $X_{attn}$ vs $X_{out}$ from the CLIP vision encoder, $N_{q} = 4$, and context length set to 12. My concern is that these design decisions/hyperparameters are tailored to the real world setting or to Libero, and whether it’s necessary to tune these to make OTTER generalize well to a different environment.

**Questions For Authors:**

1. l. 288 col. 2: What is a “primitive”?
2. What are the computational savings of using a fraction of the visual patch tokens?
3. Why is there a difference between the action horizon length and the number of predicted actions?
4. Can you describe the time limit for task termination?
5. Can you give a comparison of the number of model parameters, training time, and inference time for OTTER, Octo and OpenVLA? This is important for assessing usability.

**Relation To Broader Scientific Literature:**

OTTER offers an alternative to heavy fine-tuning of VLMs/LLMs for robotic policy learning.

**Theoretical Claims:**

N/A.

---

> ### Author Rebuttal · Authors · 2025-04-01
>
> *W1: OTTER .. performs on par with them on the synthetic Libero benchmark...*
>
> The Libero benchmark consists of simple tasks with fixed scenes, distractors, and minimal object variation. As a result, all methods can achieve comparable performance by fitting well to the benchmark’s constraints. Since CLIP is predominantly trained on real-world data, its vision encoder is more effective in real settings than in synthetic ones.
>
> In our real-world experiments, OTTER was evaluated on four primitives, unseen objects, and novel distractors. Experiment videos are available at https://ottericml.github.io/, showcasing OTTER's ability to generalize across diverse environments.
>
> *W2: Several of OTTER’s design decisions are not ablated, namely: whether to use Xattn vs Xout from the CLIP vision encoder, Nq=4, and context length set to 12.*
>
> We have included the ablation experiments on using X_out. Specifically, we trained OTTER using the X_out from CLIP on the DS-PnP and evaluated on the pick and place tasks. The results are shown in the table below:
> | Method        | Unseen Pick and Place Task Mean ± Std. Err. |
> |--------------|--------------------------------------------|
> | OTTER X_out  | 18.6% ± 4.3%                              |
> | OTTER        | 62% ± 4.2%                                |
>
> These results demonstrate that using X_attn​ significantly improves real-world performance, aligning with findings from ClearCLIP and Appendix Figure 6. Leveraging  X_attn enhances OTTER’s generalization in real-world settings compared to using X_out. We also conducted ablations in simulation to assess the impact of context length and N_q​ on performance in Libero-object tasks. The results are shown below:
>
> | Context Length | Success Rate       |
> |----------------|--------------------|
> | 4              | 90% ± 1.2%         |
> | 8              | 90% ± 1.4%         |
> | 12             | 89% ± 1.2%         |
> | 16             | 87% ± 1.3%         |
>
> | N_q | Success Rate       |
> |-----|--------------------|
> | 2   | 87% ± 1.4%        |
> | 4   | 89% ± 1.2%        |
> | 8   | 88% ± 1.3%        |
>
> Overall, our ablation studies support the idea that OTTER’s success is not solely dependent on hyperparameter tuning for a specific setting. Instead, key design choices, such as using X_attn play a more significant role in improving generalization to real-world tasks.
>
>
> *Q1*:
> We define primitive as a fundamental, reusable action or skill that can be applied across multiple tasks when combined with different objects. For example, picking up and pouring are two different primitives, Picking up an apple and picking up a banana are two different tasks but the same primitive.
>
> *Q2*:
> For an image input of size `224 × 224`, the CLIP vision encoder initially generates **257 tokens** (256 from `16 × 16` patches plus 1 CLS token). We extract `N_q = 4` text-relevant visual features from `X_attn` of size `15 d`, and concatenate them into a single token.  `d` is the latent dimension. The computational complexity of processing these features is approximately `O(4 × 15 d)`. A standard Transformer’s attention mechanism has a complexity of `O(T²d)`, where `T` is the number of input tokens. For full-resolution image tokens (`T = 257`), the computational cost is `O(257²d)`.  Thus, the ratio of computational cost when using compressed visual tokens compared to processing all 257 tokens is:  `(O(4 x15d) + O(d))/O(257^2 d)  = 0.001`. This demonstrates that our approach reduces the computational cost to approximately **0.1%** of the full-token processing cost, leading to significant efficiency gains in both training and inference.
>
> *Q3*:
> The action horizon length H=8 is the parameter used in the receding horizon control [1] for action smoothness, as in [2]. For a number of predicted actions of 12, only the first 8 numbers are used for temporal ensembles to calculate the action to execute.
>
> [1] Mayne, D. Q., & Michalska, H. (1988, December). Receding horizon control of nonlinear systems.
>
> [2] Chi, Cheng, et al. "Diffusion policy: Visuomotor policy learning via action diffusion."
>
> *Q4*:
> As in A.2, all models except OpenVLA are given a maximum of 30 seconds per trial. Due to OpenVLA's larger size and slower inference speed, it is allotted 60 seconds.
>
> *Q5*:
> Octo has 93 million parameters and an inference speed of 33 Hz. It was trained for 300k steps in 14 hours using a TPU v4-128 pod. Fine-tuning Octo on the DS-ALL dataset for 40k steps takes 6 hours using 8 A100 GPUs. OpenVLA, with 7 billion parameters and a much lower inference speed of 0.5 Hz, was trained for 14 days (21,500 A100-hours) using a cluster of 64 A100 GPUs. Fine-tuning OpenVLA on DS-ALL for 15k steps takes 15 hours on 8 A100 GPUs. OTTER, the smallest with 25.5 million parameters and the highest inference speed of 50 Hz, was pre-trained on the OXE dataset for 40k steps in 12 hours using 4 A100 GPUs. Fine-tuning on DS-ALL for 40k steps takes 6 hours on 8 A100 GPUs.

---

### Official Review · Reviewer_Yr2M · 2025-03-14

**Overall Recommendation:** 3

**Summary:**

This paper aims to address a critical issue wherein the simultaneous fine-tuning of visual and language encoders during VLA training significantly exacerbates the risk of overfitting and impairs their original perceptual capabilities. To overcome this, the authors propose an explicit alignment strategy, which utilizes language features to guide the extraction of visual features, thereby obviating the need for encoder fine-tuning. Experiments on downstream tasks demonstrate the superiority and effectiveness of this proposed method.

**Claims And Evidence:**

The experimental results presented in the paper somewhat support its claims.

**Essential References Not Discussed:**

The authors could include a discussion on related VLA works that do not freeze encoders, such as CogACT, to illustrate the potential limitations of encoder fine-tuning approaches. Specifically, methods like CogACT jointly fine-tune visual and linguistic encoders, which could lead to severe overfitting risks. This encoder fine-tuning can cause the models to lose their original perceptual and semantic generalization capabilities, making them overly specialized to specific training tasks or datasets. By analyzing such limitations, the authors could clearly demonstrate why freezing modality encoders is crucial to preserving their intrinsic representational abilities and achieving better generalization across tasks.

**Experimental Designs Or Analyses:**

In Table 2, several baseline methods, such as OpenVLA and Octo, exhibit near-zero success rates. The authors, however, did not compare their approach with more recent baseline methods like Pi-0. Thus, although the proposed method achieves noticeable improvements, the poor performance of baselines may stem from architectural limitations rather than solely from encoder-related issues.

**Methods And Evaluation Criteria:**

Intuitively, I also believe this straightforward approach can indeed yield performance improvements.

**Other Comments Or Suggestions:**

NA

**Other Strengths And Weaknesses:**

Overall, the authors present a simple yet effective method that emphasizes the necessity of preserving original encoder capabilities. This insight is valuable to the community, as it highlights a critical consideration often overlooked in current VLA research. Although the proposed method itself is straightforward and additional baseline comparisons would further strengthen the findings, I am currently inclined toward acceptance given its conceptual clarity and practical contribution.

**Questions For Authors:**

Have the authors observed task-level generalization of their proposed method, and how does it compare with contemporary approaches?

**Relation To Broader Scientific Literature:**

I believe the primary contribution of this paper is highlighting the importance of freezing the original modality encoders, rather than jointly fine-tuning them, as many concurrent VLA studies have done. Joint fine-tuning of encoders tends to induce severe overfitting, which explains why such VLA methods typically exhibit poor generalization performance. This paper introduces and empirically validates an approach that avoids fine-tuning encoders, effectively demonstrating its benefit in improving generalization.

**Theoretical Claims:**

NA

---

> ### Author Rebuttal · Authors · 2025-04-01
>
> *The authors, however, did not compare their approach with more recent baseline methods like Pi-0.*
>
> We thank the reviewer for pointing this out. We have added experiments comparing OTTER with Pi-0 as below.
> | Method | Pouring | Drawer | Poking | Pick and Place | Mean±Std. Err. |
> |-----------|-----------|----------|----------|----------|----------|
> | π0-Fast-Droid | 0% | 0% | 0% | 61% | 29% ± 3.5% |
> | Finetuned π0-Fast-Droid | 0% | 45%|  27% | 51% | 35% ± 3.8% |
> | OTTER-OXE-L | 77% | 75% | 93% | 75% | 77% ± 3.3% |
>
> Specifically, we consider the Pi-0 Fast [1] model trained on Droid. Since our experiment setup is the droid setup, we first evaluate the pi-0 performance directly on our setup, denoted as $\pi_0$-Fast-Droid. As shown in the table, while $\pi_0$-Fast-Droid achieves decent performance on the pick and place primitives, it fails on all the other three primitives as the majority of the Droid dataset is the pick and place primitive. Therefore, we also consider fine-tuning $\pi_0$-Fast-Droid on the DS-ALL, denoted as Finetuned $\pi_0$-Fast-Droid. Finetuned $\pi_0$-Fast-Droid achieves non-zero success rate on Drawer and Poking primitives, but still fails on the pouring primitive. There is also a degraded performance on the Pick and Place primitives for Finetuned $\pi_0$-Fast-Droid. This highlights the difficulty of our experiment setup. OTTER-OXE-L can achieve high success rate on all four primitives on the same amount of demonstrations, indicating that using text-aware visual features extracted from a pre-trained VLM can increase the data efficiency and enhance the generalization ability.
>
> [1] Pertsch, Karl, Kyle Stachowicz, Brian Ichter, Danny Driess, Suraj Nair, Quan Vuong, Oier Mees, Chelsea Finn, and Sergey Levine. "Fast: Efficient action tokenization for vision-language-action models." arXiv preprint arXiv:2501.09747 (2025).
>
> *I noticed that the tasks evaluated are predominantly simple pick-and-place scenarios.*
>
> We have primitives other than the pick-and-place as shown in Table 2, where we consider 4 primitives, pouring, drawer, poking and pick-and-place. We have provided the videos for the experiments on https://ottericml.github.io/ to show the diversity of the evaluation tasks.
>
> *The authors could include a discussion on related VLA works that do not freeze encoders, such as CogACT.*
>
> We thank the reviewer for pointing out this reference. We appreciate and agree with the reviewer’s discussion on CogACT and will include this in the related work of the revised draft.
>
> *Have the authors observed task-level generalization of their proposed method, and how does it compare with contemporary approaches?*
>
> In OTTER, we define primitive as a fundamental, reusable action or skill that can be applied across multiple tasks when combined with different objects. For example, picking up and pouring are two different primitives, picking up an apple and picking up a banana are two different tasks but the same primitive. OTTER has shown strong task-level generalization as shown in Table 1 and 2. We assume the reviewer is referring to primitive-level generalization.
>
> As the reviewer pointed out earlier, keeping a frozen pre-trained Visual-Language Model (VLM) primarily enhances visual generalization. In OTTER, our focus is on leveraging this strength to improve object-level generalization. While OTTER does not explicitly target primitive-level generalization, it can bring certain benefits in enabling it. By reducing the burden of generalizing to unseen objects, the model can allocate more capacity to learning task primitives and compositional behaviors, which enables more efficient adaptation to novel primitives. We agree that exploring primitive-level generalization is a compelling future direction, and we explicitly encourage work in this area.

---

### Official Review · Reviewer_8GXo · 2025-03-14

**Overall Recommendation:** 3

**Summary:**

OTTER is a Vision-Language-Action (VLA) model that enhances robotic task execution by leveraging the semantic alignment capabilities of pre-trained Vision-Language Models (VLMs) without fine-tuning. By extracting text-aware visual features aligned with task instructions, OTTER preserves the rich visual-language understanding from pre-training, enabling effective handling of novel objects and environments. It demonstrates strong zero-shot generalization, outperforming state-of-the-art VLA models on unseen robot manipulation tasks. Empirical results show that OTTER's performance scales with larger pre-trained vision-language encoders, increased policy network capacity, and pre-training on larger robot datasets, making it a robust solution for diverse and unseen robotic scenarios.

## Update after rebuttal
After reading all reviewers' comments and authors' responses, some of my confusion about technical details are explained. However, in comparison with previous work in LLaVA-style, author select only 1 token, such as CLS token, to keep the same with OTTER's setting, but it's not the setting of OpenVLA, so that I think there is a promblem of comparing with OpenVLA.
OpenVLA builds their framework based on large language model such as LLaMA2, but OTTER is based on a Transformer-based policy model. Although OTTER has better performance than OpenVLA in experiments, OTTER's capacity of dealing with in-the-wild instructions are questionable, and maybe OpenVLA is not suitable to be a baseline of OTTER.

Overall, OTTER designed a novel framework to extract and merge multi-modal features in VLA tasks efficiently and obtained an impressive performance in several tasks. Therefore, I will keep my rating unchanged.

**Claims And Evidence:**

Otter use a novel parameter-efficient text-aware vision feature extraction scheme in VLA model, rather than commonly used finetuning or projection. By selectively use visual features, visual information can be extracted and used efficiently. However, experiments and ablation studies mainly focus on effects of embodied encoder or pre-trained CLIP encoder. Effectiveness of text-aware visual feature extraction are not well explored.

**Essential References Not Discussed:**

N/A

**Experimental Designs Or Analyses:**

Experiments presented by this work is quite impressive, which includes environments both in simulation and real world. Also, generalization ability on unseen tasks are well tested. But in my opinion, ablation study lack of the analysis of claimed text-aware visual feature extraction. Details of DFP-OTTER can be shown and compared in supplementary.
Meanwhile, mainstream VLMs, such as LLaVA, usually freezes CLIP encoder during training rather than finetuning. Another baseline using linearly projected frozen CLIP features shall be included.

**Methods And Evaluation Criteria:**

The method part mainly focus on visual feature processing to solve problems in utilizing CLIP features in existing VLMs. Although the improvement in visual features are reasonable, results in Table.1 shows that embodied features have a large effect on overall performance. Thus, it's quite confusing whether the visual features are the key in this kind of tasks.

**Other Comments Or Suggestions:**

See above.

**Other Strengths And Weaknesses:**

Strength:
Otter adopts current VLM paradigm and explored into VLA model. By designing a selective visual feature extraction scheme, Otter achieves impressive exprimental results.

Weakness:
1. Baseline of linearly projected CLIP features shall be compared.
2. Beyond text-aware visual feature extraction, visual features are further compressed before input into policy network. Details of this step are not well presented and analysed.

**Questions For Authors:**

N/A

**Relation To Broader Scientific Literature:**

Otter achieves impressive performance both in simulation and real world environments, trained and unseen tasks. The benchmarking setting is quite important in the field. And it highlights the importance of well-pretrained vision encoder in VLA models.

**Theoretical Claims:**

N/A

---

> ### Author Rebuttal · Authors · 2025-04-01
>
> Q1: "Effectiveness of text-aware visual feature extraction are not well explored."
>
> We agree that explicitly isolating the impact of our proposed text-aware visual feature extraction is important. The ablation study (DFP-OTTER baseline in Tables 1 and 3 and Appendix D, Table 9) was designed to illustrate the benefit of explicitly selecting text-aware visual tokens compared to directly passing visual features to the policy transformer. Specifically, the results demonstrate that selective text-aware feature extraction substantially outperforms this direct-feature-passing baseline on unseen scenarios.
>
> Q2: "Although the improvement in visual features are reasonable, results in Table.1 shows that embodied features have a large effect on overall performance. Thus, it's quite confusing whether the visual features are the key in this kind of tasks."
>
> While embodied features (proprioceptive information such as robot end-effector pose) significantly contribute to overall performance—given that robotic tasks inherently require spatial grounding—the visual features remain fundamentally critical. If the policy is provided with only embodied features without meaningful visual inputs, the robot cannot perceive objects at all, yielding a zero success rate. Furthermore, pi0 also takes the embodied features. As shown in the additional results in response to reviewer Yr2M, OTTER outperforms pi0, indicating the significance of text-aware visual features.
>
> Q4/W1: "Mainstream VLMs, such as LLaVA, usually freeze CLIP encoder during training rather than finetuning. Another baseline using linearly projected frozen CLIP features shall be included."
>
> Thank you for raising this important point. We address this from two angles:
>
> 1. To investigate the effectiveness of simply using frozen CLIP features with linear projection, we conducted a specific ablation using the CLS token from the CLIP encoder as the only visual representation (see Appendix D, Table 9). For convenience, we list the relevant results here:
> | Method                      | Unseen Tasks |
> |-----------------------------|--------------|
> | OTTER (CLS token only DFP)  | 6% ± 0.8%    |
> | OTTER (Ours)                | 62% ± 4.2% |
>
> These results show that using just the CLS token with a linear projection under the OTTER formulation—similar to mainstream frozen-encoder approaches like LLaVA—is insufficient for achieving generalization in robotic manipulation tasks.
>
> 2. Furthermore, unlike mainstream VLM architectures such as LLaVA, prior state-of-the-art VLA models (e.g., OpenVLA) explicitly demonstrate that fine-tuning the visual encoder significantly improves robotic performance. As reported by OpenVLA (Section 3.4 and Table 1), fine-tuning the vision encoder leads to notably higher success rates compared to using a fully frozen vision encoder. This suggests that directly adopting the mainstream frozen-encoder approach from general VLM literature is not optimal for robotic applications, and our explicit text-aware feature extraction provides an effective alternative solution.
> We will clearly articulate both points in our revised manuscript to thoroughly address this comparison.
>
> W2: "Beyond text-aware visual feature extraction, visual features are further compressed before input into policy network. Details of this step are not well presented and analyzed."
>
> Thank you for pointing this out. We compress visual features using a learnable attention pooling operation to produce a compact representation, as briefly described in Section 3.2. Specifically, the text-aware visual feature extraction step outputs multiple (m) text-aligned visual tokens (empirically, we used the first 15 text-tokens, which sufficiently covers the instructions) each of dimension 768. Directly concatenating these tokens would result in a prohibitively high-dimensional input (15 × 768) for the policy transformer, significantly increasing computational complexity. Therefore, to mitigate this issue and maintain computational efficiency, we introduce a learnable attention pooling layer to aggregate these visual tokens to reduce the dimension before passing them into the policy. This attention pooling layer has N_q = 4 query tokens and a latent dimension of 64. We further concatenated these 4 tokens into 1 token.
> We will clarify and expand this step in the revised manuscript, explicitly discussing both the motivation behind the dimensionality reduction and the details of the attention pooling mechanism.

---

### Official Review · Reviewer_XXz9 · 2025-03-23

**Overall Recommendation:** 2

**Summary:**

The OTTER keeps pre-trained VLMs fixed to preserve the rich semantic understanding acquired during pretraining, enabling strong zero-shot generalization to novel objects and environments, as demonstrated in both simulation and real-world experiments. This text-aware visual feature is extracted using pre-trained CLIP on the top of the VLM. The experiments are conducted on a real robot and Libero with comparison to Octo and OpenVLA.

**Claims And Evidence:**

Although OTTER claims to "preserve the generalizability of VLMs for effective performance in unseen scenarios," the definition of "unseen scenarios" remains unclear.

**Essential References Not Discussed:**

The central idea behind Otter is to leverage text-aware visual features to enhance the effectiveness of VLA. However, several existing VLA models—for example, pi0—have not been discussed. Additionally, pertinent works that employ a chain-of-thought approach in VLA, such as embodied CoT and CoT-VLA, are also omitted. Given that using text-aware visual features contrasts with these methods, it is crucial to include them in the discussion.

**Experimental Designs Or Analyses:**

1) The baseline DFP-Otter is not a direct comparison with the base VLA. Typical VLA models use a pre-trained VLM, such as OpenVLA, and do not employ an attention pooling layer to obtain independent tokens. A fairer comparison would involve using the same OpenVLA architecture and simply adding text-aware visual features to demonstrate effectiveness, rather than building an entirely different model.

2) The DFP does not appear to be a plausible architecture for passing visual features to a transformer. Other common methods in VLM—such as using an MLP (LLaVA), q-former (BLIP), c-abstractor (CVPR'24), or feature pyramid extractor (DVCP, CVPR'24)—should be considered.

3) While Otter is claimed to handle many "unseen tasks," the tasks presented in the appendix (real-world tasks) mainly involve attributes like object color or objects already seen during training. Since CLIP can effectively handle these cases, it would be more informative to assess its performance on truly unseen objects—for example, whether it can preserve correct visual tokens.

4) The supplementary material does not include video, which leaves unclear how challenging the tasks are and whether the evaluation is robust.

5) Although CLIP is used as a prior for selecting text-aware visual tokens, other methods, such as VLM, could serve the same function. A comparison with VLM would add value to the study.

6) Libero represents a relatively simple simulation task. Given the emphasis on "unseen tasks" in this work, alternative simulations—such as a language table—might be more suitable for evaluation.

**Methods And Evaluation Criteria:**

The method makes sense, but the experiments are problematic; further details are provided in the following section.

**Other Comments Or Suggestions:**

N/A

**Other Strengths And Weaknesses:**

Other Strengths:
The paper is well-written, and its idea is both straightforward and intuitively effective.

Weaknesses:
The experimental results are underwhelming. As mentioned earlier, OTTER is highly dependent on the reliability of CLIP, and it would be beneficial to see experiments conducted in more complex real-world scenarios.

**Questions For Authors:**

N/A

**Relation To Broader Scientific Literature:**

Vision-language-action models have emerged as a popular domain in imitation learning for robotics. This paper presents a straightforward technique to enhance VLA models.

**Theoretical Claims:**

N/A

---

> ### Author Rebuttal · Authors · 2025-04-01
>
> **Claims and Evidence**
>
> Q1: Clarification on "unseen scenarios"
>
> Thank you for raising this point. In our paper, "unseen scenarios" explicitly refer to tasks involving entirely novel objects or novel combinations of objects and spatial configurations not encountered during training. Crucially, our evaluation—both in real-world (Section 4.1, Appendix A.2) and LIBERO simulation benchmarks—requires the model to distinguish the correct target object from visually similar distractors. Our tasks indeed evaluate visual grounding generalization based on language instructions.
>
> **Experimental Designs and Analyses**
>
> Q1: Regarding the fairness of baseline comparisons (DFP-OTTER vs. OpenVLA)
>
> We appreciate this concern and clarify our intentions:
> 1. Purpose of DFP-OTTER Baseline: it demonstrates explicitly that directly passing visual/textual features into a vanilla transformer is insufficient, underscoring the necessity of selective text-aware visual feature extraction for generalization.
>
> 2. Compatibility with existing VLM-based VLAs: Our paper introduces a novel VLA architecture (OTTER) specifically designed around text-aware visual feature extraction. Adding such features directly into existing VLM-based models like OpenVLA would necessitate substantial re-training of the base VLM (e.g., Prismatic VLM [1]), exceeding our available computational resources. Exploring this integration is indeed a compelling future research direction, which we explicitly encourage.
>
> [1] Karamcheti, Siddharth, et. al. "Prismatic VLMs: Investigating the design space of visually-conditioned language models." ICML 2024.
>
> Q2: Regarding plausibility of Direct Feature Passing (DFP)
>
> We included the Direct Feature Passing (DFP) baseline specifically as a minimal ablation to clearly demonstrate the benefit of selective, text-aware visual feature extraction. Introducing additional intermediate methods like MLPs or Q-formers within the VLM would divert focus from our primary contribution, which is explicitly leveraging pre-trained CLIP for improved generalization.
>
> Q3: Clarification on experiments involving truly unseen objects
>
> Our experimental setup explicitly uses entirely novel objects, as detailed in Appendix A.2, Table 6. These unseen objects differ not only in color but also in geometry and texture. Each evaluation includes explicit distractors, ensuring robust assessment of the model’s visual-language grounding capability. OTTER's superior performance demonstrates effective grounding on genuinely novel objects.
>
>
> Q4: Availability of supplementary video material
>
> Supplementary videos demonstrating task complexity and model performance are now available at our anonymous website: https://ottericml.github.io/
>
> Q5: CLIP vs. other VLM alignment methods
>
> While existing VLM-based approaches (e.g., OpenVLA, Pi0-Fast) **implicitly** learn alignments between text and visual tokens through fine-tuning their respective large-scale pre-trained models (such as Prismatic or PaliGemma), our method **explicitly** formulates text-aware visual token extraction as a separate, parameter-free process leveraging the frozen pre-trained CLIP model.
> Thus, the comparison requested by the reviewer is already covered by our existing baseline evaluations against state-of-the-art methods such as OpenVLA (Tables 1–3) and Pi0-Fast (added in this rebuttal). These comparisons illustrate the advantage of the explicit, selective token extraction approach compared to implicit token alignment methods embedded in other VLM-based approaches.
>
> Q6: Suitability of LIBERO for simulation evaluation
>
> We selected LIBERO explicitly due to its usage in recent state-of-the-art VLA works (OpenVLA, π0-Fast-Droid), facilitating fair and direct comparisons. LIBERO supports evaluations involving explicit distractors under varied language instructions, thus providing meaningful assessments of visual-language generalization.
>
> **Essential References Not Discussed**
>
> Q1: Discussion of embodied CoT and CoT-VLA works
>
> We appreciate this suggestion and will explicitly include and contrast these methods in the updated related work section of the revised manuscript.
>
> Q2: Additional comparison with π0 / its extensions
>
> We include additional comparisons and discussions with π0-Fast-Droid in response to reviewer Yr2M, the current state-of-the-art VLA model, leveraging extensive pre-training on proprietary robot data.
>
> **Weaknesses**
>
> W1: Dependence on CLIP and complexity of real-world scenarios
>
> We agree that evaluating more complex real-world scenarios is valuable. However, current evaluations (please see updated tables on the website) already demonstrate significant challenges for state-of-the-art models (OpenVLA, Octo, π0-Fast-Droid), validating our approach’s robustness. Exploring tasks with more complex interactions remains an important future direction, explicitly mentioned in the revision.

---

### Decision · Program_Chairs · 2025-05-01

**Decision:**

Accept (poster)

**Comment:**

This paper introduces OTTER, a novel Vision-Language-Action (VLA) model that addresses the issue of fine-tuning pre-trained vision-language models (VLMs) for robotic tasks. OTTER aims to preserve the rich semantic understanding of VLMs by extracting text-aware visual features without requiring fine-tuning, achieving strong zero-shot generalization capabilities across novel objects and environments.

Reviewers have highlighted several strengths, including the novelty of the method and the promising real-world experimental results. The approach of leveraging pre-trained CLIP features for selective, task-relevant visual feature extraction is considered a key contribution, especially in light of its efficiency gains and strong performance on unseen tasks.


Overall, the paper offers a compelling approach with strong results but could be improved with additional experiments and a more thorough exploration of some technical details. The AC agrees with the reviewers to accept the submission.